# Infection prevention and control compliance among exposed healthcare workers in COVID-19 treatment centers in Ghana: A descriptive cross-sectional study

Mary Eyram Ashinyo[1]*, Stephen Dajaan Dubik[2], Vida Duti[3], Kingsley Ebenezer Amegah[4], Anthony Ashinyo[5], Brian Adu Asare[6], Angela Ama Ackon[7], Samuel Kaba Akoriyea[1], Patrick Kuma-Aboagye[8]

1 Institutional Care Division, Ghana Health Service Headquarters, Accra, Ghana, 2 School of Allied Health Sciences, University for Development Studies, Tamale, Ghana, 3 IRC-Ghana, Cantonments, Accra, Ghana, 4 Department of Health Information, Hohoe Municipal Hospital, Hohoe, Ghana, 5 National AIDS/STI Control Programme, Accra, Ghana, 6 Directorate of Pharmacy, Ministry of Health, Accra, Ghana, 7 World Health Organisation, Ghana Country Office, Accra, Ghana, 8 Office of the Director-General, Ghana Health Service Headquarters, Accra, Ghana

* mary.ashinyo@ghsmail.org

**Data Availability Statement:** All relevant data are within the paper and its Supporting Information files.

**Funding:** We confirm receipt of funding from IRC-Ghana. The funders had no role in study design, data collection and analysis, decision to publish, or preparation of the manuscript. MEA, IRC-Ghana https://www.ircwash.org/ghana

## Abstract

Compliance with infection prevention and control (IPC) protocols is critical in minimizing the risk of coronavirus disease (COVID-19) infection among healthcare workers. However, data on IPC compliance among healthcare workers in COVID-19 treatment centers are unknown in Ghana. This study aims to assess IPC compliance among healthcare workers in Ghana's COVID-19 treatment centers. The study was a secondary analysis of data, which was initially collected to determine the level of risk of COVID-19 virus infection among healthcare workers in Ghana. Quantitative data were conveniently collected using the WHO COVID-19 risk assessment tool. We analyzed the data using descriptive statistics and logistic regression analyses. We observed that IPC compliance during healthcare interactions was 88.4% for hand hygiene and 90.6% for Personal Protective Equipment (PPE) usage; IPC compliance while performing aerosol-generating procedures (AGPs), was 97.5% for hand hygiene and 97.5% for PPE usage. For hand hygiene during healthcare interactions, lower compliance was seen among nonclinical staff [OR (odds ratio): 0.43; 95% CI (Confidence interval): 0.21–0.89], and healthcare workers with secondary level qualification (OR: 0.24; 95% CI: 0.08–0.71). Midwives (OR: 0.29; 95% CI: 0.09–0.93) and Pharmacists (OR: 0.15; 95% CI: 0.02–0.92) compliance with hand hygiene was significantly lower than registered nurses. For PPE usage during healthcare interactions, lower compliance was seen among healthcare workers who were separated/divorced/widowed (OR: 0.08; 95% CI: 0.01–0.43), those with secondary level qualifications (OR 0.08; 95% CI 0.01–0.43), non-clinical staff (OR 0.16 95% CI 0.07–0.35), cleaners (OR: 0.16; 95% CI: 0.05–0.52), pharmacists (OR: 0.07; 95% CI: 0.01–0.49) and among healthcare workers who reported of insufficiency of PPEs (OR: 0.33; 95% CI: 0.14–0.77). Generally, healthcare workers' infection prevention and control compliance were high, but this compliance differs across the different groups of health

**Competing interests:** The authors have declared that no competing interests exist.

professionals in the treatment centers. Ensuring an adequate supply of IPC logistics coupled with behavior change interventions and paying particular attention to nonclinical staff is critical in minimizing the risk of COVID-19 transmission in the treatment centers.

## Introduction

The COVID-19 Pandemic, which emanated from Wuhan, China, has devastated the global community, disrupting all aspects of human lives [1, 2]. As of 8 February 2021, there were 105,805,951 reported COVID-19 cases with 2,312,278 deaths globally [1]. Currently (February 5 2021), there are 72,328 confirmed COVID-19 cases in Ghana, with 472 reported deaths and 6,707 active cases [3]. The disease is a highly infectious viral respiratory disease that is more severe in older people and people with underlying medical conditions [4, 5]. COVID-19 infection can either be asymptomatic or symptomatic with prominent ones being fever, cough, sore throat and shortness of breath [6, 7].

Healthcare workers play a critical role in fighting the COVID-19 pandemic and are at greater risk of COVID-19 virus infection in the line of duty [8]. For instance, data from recent studies showed healthcare workers are more likely to be exposed to SARS-COV-2 [9] and are, therefore, at higher risk of COVID-19 infection than the general community [10]. Hence, the impact of the COVID-19 pandemic on healthcare workers has been enormous [11]. However, prevention remains the best weapon for protecting healthcare workers against the COVID-19 pandemic [12]. Therefore, adherence to infection prevention and control protocols is critical at minimizing healthcare workers exposure to the severe acute respiratory syndrome coronavirus 2 (SARS-CoV-2) [8, 13]. Indeed, correct and consistent compliance with IPC protocols is effective in minimizing the risk of COVID-19 infection [8, 10]. Compliance with IPC protocols is facilitated by training of healthcare workers on IPC, provision of IPC materials and regular audit of IPC practices [14]. Generally, IPC strategies in response to highly infectious diseases, such as COVID-19, should include early recognition, physical distancing, source control, taking precautions and appropriate use of PPEs, restriction of movement, environmental cleaning and disinfection as well as support for healthcare workers [14, 15].

COVID-19 infection among frontline healthcare workers put patients, other healthcare workers and the general community at risk of infection. Minimizing exposure of healthcare workers to the SARS-CoV-2 is the best option for protecting frontline healthcare workers from COVID-19 infection, and this is best done through healthcare worker adherence with IPC protocols as well as inoculating against the SARS-COV-2 [16]. Therefore, an understanding of IPC compliance among healthcare workers managing COVID-19 patients is essential for preventing the spread of COVID-19 infections among healthcare workers, reducing secondary transmission in the treatment centers and updating IPC policies in Ghana.

## Materials and methods

The study was a descriptive cross-sectional survey that was conducted from May to August 2020 in four COVID-19 treatment centers located in Greater Accra (Pentecost Convention Center, Ga East Municipal Hospital) and Ashanti regions (Barekese and Kumasi South Hospital). The Greater Accra and Ashanti regions were purposively selected, being epicenters, and having the greatest burden of cases, with 50% and 24% of all cases in Ghana, respectively.

## Study participants, sample size and sampling

The study participants included clinical and nonclinical healthcare workers working in COVID-19 treatment centers. The sample size was calculated using the Cochran formula [17]; $N = \frac{z^2 \times p(1-p)}{d^2}$ **at** 95% confidence interval (1.96), exposure level of 50% (0.5), and a 5% margin of error (0.05). After adding a contingency of 10%, the estimated sample size was 424. Using convenience sampling, healthcare workers were invited to participate in the study. Those who consented to participate were given the questionnaire with clear instruction on how to complete the questionnaire. Completed questionnaire was collected immediately. Out of 424 questionnaires distributed, 408 were completed and retrieved. Up to 328 out of the 408 respondents were exposed to a COVID-19 patient and were, therefore, qualified to answer questions on adherence to IPC measures during healthcare interactions with COVID-19 patients. Of the 328, eighty of them performed AGP. Compliance with IPC measures when performing AGP was therefore assessed for only 80 of the study participants. The details of the methodology have been explained elsewhere [18].

## Ethical consideration

The study was part of a large study titled "Exposure risk Assessment: A survey among frontline healthcare workers in designated COVID-19 treatment centers", which was approved by the Ghana Health Service Ethics Review Committee. Written informed consent was obtained from all study participants.

## Study variables

We adopted the study tool from the WHO risk assessment tool for healthcare workers in the context of COVID-19 [19], attached as S1 File. This tool was used to assess the healthcare workers reported compliance with IPC measures during healthcare interactions and when performing AGPs on COVID-19 patient. Ten (10) items each, were assessed for compliance to IPC measures during healthcare interactions and while performing AGPs, (Tables 1 and 2) with Likert responses: "always, as recommended", "most of the time", "occasionally" and "rarely". Healthcare workers were scored one (compliant) if the healthcare worker responded either "always, as recommended" or "most of the time", otherwise the healthcare worker was scored zero (noncompliant) (Tables 1 and 2).

## Statistical analysis

We analyzed the data using STATA 14.2. First, descriptive statistics were used to present the study participant characteristics, compliance with hand hygiene and PPE usage in text, figures and tables. Overall compliance during healthcare interactions with COVID-19 patients and when performing AGPs were also summarized using text and figures. Logistic regression analyses were performed to ascertain the association between healthcare workers' sociodemographic information, availability of IPC facilities and IPC compliance during healthcare interactions with COVID-19 patients. All variables were significant at p-values less than 0.05 at 95% confidence intervals.

# Results and discussion

## Results

**Sociodemographic information of the study participants and the availability of IPC facilities.** The sociodemographic characteristics of the healthcare workers are presented in

**Table 1. Measure of infection prevention and control compliance during healthcare interactions.**

| PPE usage domain | Measure of compliance |
|---|---|
| Single-use gloves | Healthcare worker responds either "always as recommended" or "most of the time." |
| Medical mask | As seen above |
| Face shield or goggles/protective glass | As seen above |
| Disposal gown | As seen above |
| Remove and replace PPE according to protocol | As seen above |
| **Hand hygiene domain** | **Measure of compliance** |
| Perform hand hygiene before and after touching COVID-19 patient | Healthcare worker responds either "always as recommended" or "most of the time." |
| Perform hand hygiene before and after any clean or aseptic procedure | As seen above |
| Perform hand hygiene after exposure to body fluids | As seen above |
| Perform hand hygiene after touching patient surroundings | As seen above |
| Frequent decontamination of high touch surfaces | As seen above |
| Compliance with IPC during healthcare interactions with COVID-19 patients | Healthcare worker responds either "always as recommended" or "most of the time." to all variables on PPE use and hand hygiene domains |

Table 3. The average age of the healthcare workers in this study was 33 years, with most (62.2%) of them between the age brackets 30–49. More (70.7%) healthcare workers from the Greater Accra region participated in the study compared to the Ashanti region. Most (56.4%) of the study participants were females, 50.6% of the healthcare workers were married, and the majority (32.6%) of the healthcare workers had certificate as their highest qualification. Clinical staff formed the majority (78.0%) of the healthcare workers. Participating health professionals were

**Table 2. Measure of infection prevention and control compliance when performing AGPs on COVID-19 patients.**

| PPE use domain | Measure of compliance |
|---|---|
| Single-use gloves | Healthcare worker responds either "always as recommended" or "most of the time." |
| N95 mask or respirator equivalent | As seen above |
| Face shield or goggles/protective glass | As seen above |
| Disposal gown | As seen above |
| Remove and replace PPE according to protocol | As seen above |
| **Hand hygiene domain** | **Measure of compliance** |
| Perform hand hygiene before and after touching COVID-19 patient | Healthcare worker responds either "always as recommended" or "most of the time." |
| Perform hand hygiene before and after any clean or aseptic procedure | As seen above |
| Perform hand hygiene after exposure to body fluids | As seen above |
| Perform hand hygiene after touching patient surroundings | As seen above |
| Frequent decontamination of high touch surfaces (at least three times) | As seen above |
| Compliance with IPC measures when performing AGP on COVID-19 patients | Healthcare worker responds either "always as recommended" or "most of the time." to all variables on PPE use and hand hygiene domains |

**Table 3. Sociodemographic information of the study participants.**

| VARIABLES | Frequency | Percent (%) |
|---|---|---|
| **Healthcare worker characteristics (n = 328)** | | |
| Region | | |
| Ashanti | 96 | 29.3 |
| Greater Accra | 232 | 70.7 |
| Mean age (SD), Min–Max | | 32.6 (6.14), 20–49 |
| Age (In years) | | |
| < 30 | 124 | 37.8 |
| 30–49 | 204 | 62.2 |
| Gender | | |
| Female | 185 | 56.4 |
| Male | 143 | 43.6 |
| Marital Status | | |
| Single | 156 | 47.6 |
| Married | 166 | 50.6 |
| Separated/Divorced/Widowed | 6 | 1.8 |
| Highest Qualification | | |
| Secondary level qualification | 35 | 10.7 |
| Certificate* | 107 | 32.6 |
| Diploma | 81 | 24.7 |
| Bachelor | 82 | 25.0 |
| Masters | 23 | 7.0 |
| Staff Category | | |
| Nonclinical | 72 | 22.0 |
| Clinical | 256 | 78.0 |
| Type of health professional | | |
| Assistant nurse or equivalent | 45 | 13.7 |
| Cleaner | 37 | 11.3 |
| Laboratory personnel | 10 | 3.0 |
| Medical doctor | 26 | 7.9 |
| Midwife | 24 | 7.3 |
| Registered nurse | 143 | 43.6 |
| Pharmacist | 6 | 1.8 |
| Other staff** | 37 | 11.3 |
| Work Experience | | |
| < 5 | 167 | 50.9 |
| 5–10 | 104 | 31.7 |
| 11+ | 57 | 17.4 |
| **Availability of IPC facilities** | | |
| Experienced an interruption in water supply | | |
| No | 286 | 87.2 |
| Yes | 42 | 12.8 |
| Sufficiency of PPEs | | |
| No | 44 | 13.4 |
| Yes | 284 | 86.6 |
| Training on IPC | | |
| No | 7 | 2.1 |

*(Continued)*

**Table 3.** (Continued)

| VARIABLES | Frequency | Percent (%) |
|---|---|---|
| Yes | 321 | 97.9 |

\*Healthcare workers who pursued certificate program as their highest qualification;

\*\*Other staff included Physical and respiratory therapist, catering staff, Admission/reception clerks, administrative and IT manager, Clinical engineers;

\*\*\* Nonclinical staff included cleaners, catering staff, administrative and IT managers,

registered nurses (43.6%), assistant nurses (13.7%), cleaners (11.3%), medical doctors (7.9%), midwives (7.3%), laboratory personnel (3.0%), pharmacists (1.8%) and other health professionals (11.3%). The majority of the HCWs indicated that they have never experienced an interruption in water supply in the treatment centers (87.2%), that PPEs were sufficient (86.6%) and a significant proportion (97.9%) of the healthcare workers have received training on IPC (Table 3).

**Infection prevention and control reported compliance during healthcare interactions with COVID-19 patients.** Compliance with hand hygiene during healthcare interactions with COVID-19 patients was high (88.4%). Similarly, compliance with PPEs usage during healthcare interactions with COVID-19 patients was high (90.6%) (Fig 1). Detailed analysis showed that adherence with frequent decontamination of high touch surfaces and hand hygiene before and after touching COVID-19 patients was high (97.3%). The healthcare workers also reported performing hand hygiene after touching patient surroundings (96.3%), after exposure to body fluids (95.1%), before and after any clean or aseptic procedure (93.9%) (Fig 1).

Compliance with medical mask use was nearly universal (98.8%). Disposable gown use was the lowest (93.9%) complied in PPE usage domain. Compliance with single-use gloves was

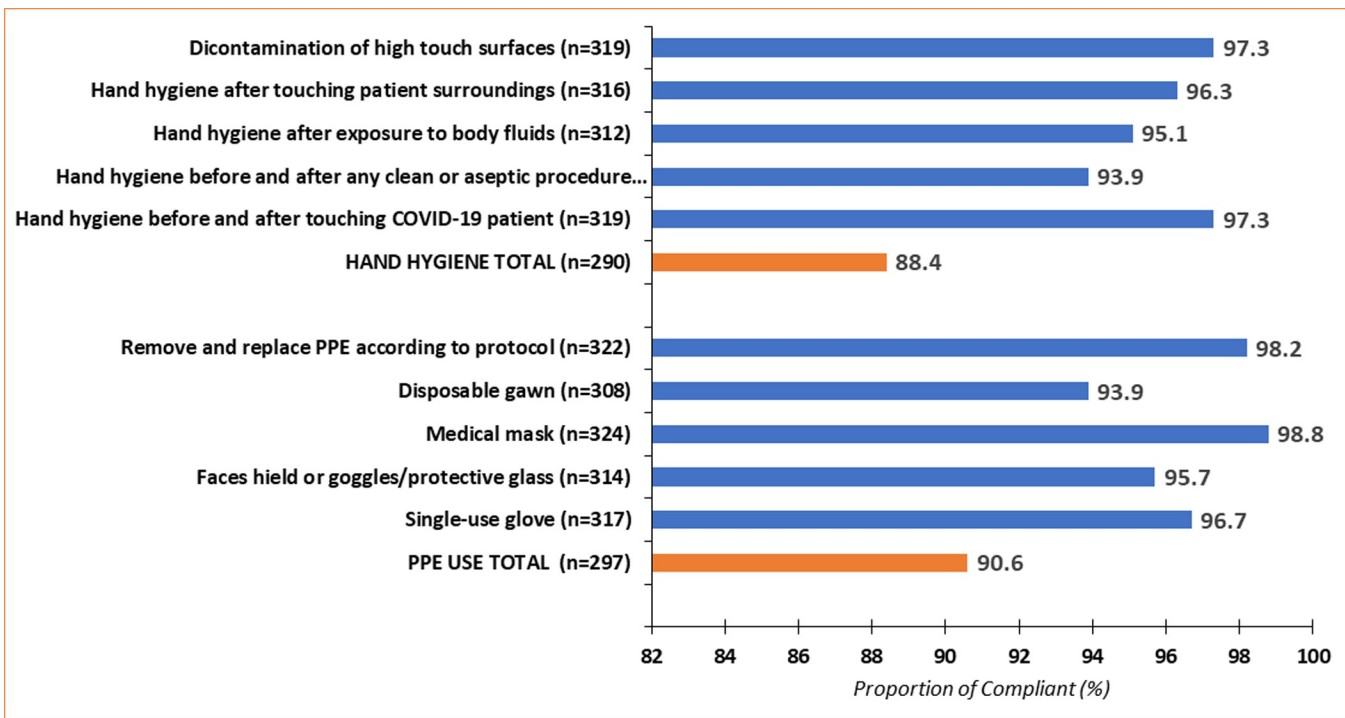

**Fig 1. Infection prevention and control reported compliance during healthcare interactions with COVID-19 patients.**

96.7%, and 98.2% of the healthcare workers reported compliance with removing and replacing PPE according to protocol (Fig 1). Details analysis on IPC compliance during healthcare interactions is attached as S1 Table.

**Infection prevention and control reported compliance when performing aerosol-generating procedures on a COVID-19 patient.** Compared to hand hygiene and PPE usage compliance during healthcare interactions, healthcare workers reported the same compliance with hand hygiene (97.5%) and PPE usage (97.5%) when performing AGPs (Fig 2). Compliance with hand hygiene after touching the patient surroundings, before and after any clean or aseptic procedure was universal (100%). Healthcare workers' compliance with N95 respirator use, disposable gown, face shield or goggles/protective glass and glove use were also universal when performing AGPs (Fig 2). Details analysis on IPC compliance during AGPs is attached as S2 Table.

**Total compliance during healthcare interactions and when performing AGPs.** Compliance with IPC during healthcare interactions was high (80.8%). However, the highest (95%) compliance with IPC was when performing AGPs (Fig 3).

**Association between healthcare workers' sociodemographic information, availability of IPC facilities and IPC compliance during healthcare interaction with a COVID-19 patient.** Table 4 shows an association between healthcare workers' sociodemographic information, availability of IPC facilities and IPC compliance during healthcare interaction with a COVID-19 patient. Region of residence, age, gender, work experience, interruption in water supply and IPC training was not associated with compliance in either PPE use or hand hygiene. Risk factors for lower compliance with PPE use were being separated/divorced/widowed (OR 0.08; 95% CI 0.01–0.43), having secondary level qualifications (OR 0.08; 95% CI 0.01–0.43) and being a non-clinical staff (OR 0.16 95% CI 0.07–0.35). We also observed lower odds of compliance with PPEs usage among cleaners (OR 0.16; 95% CI 0.05–0.52) and

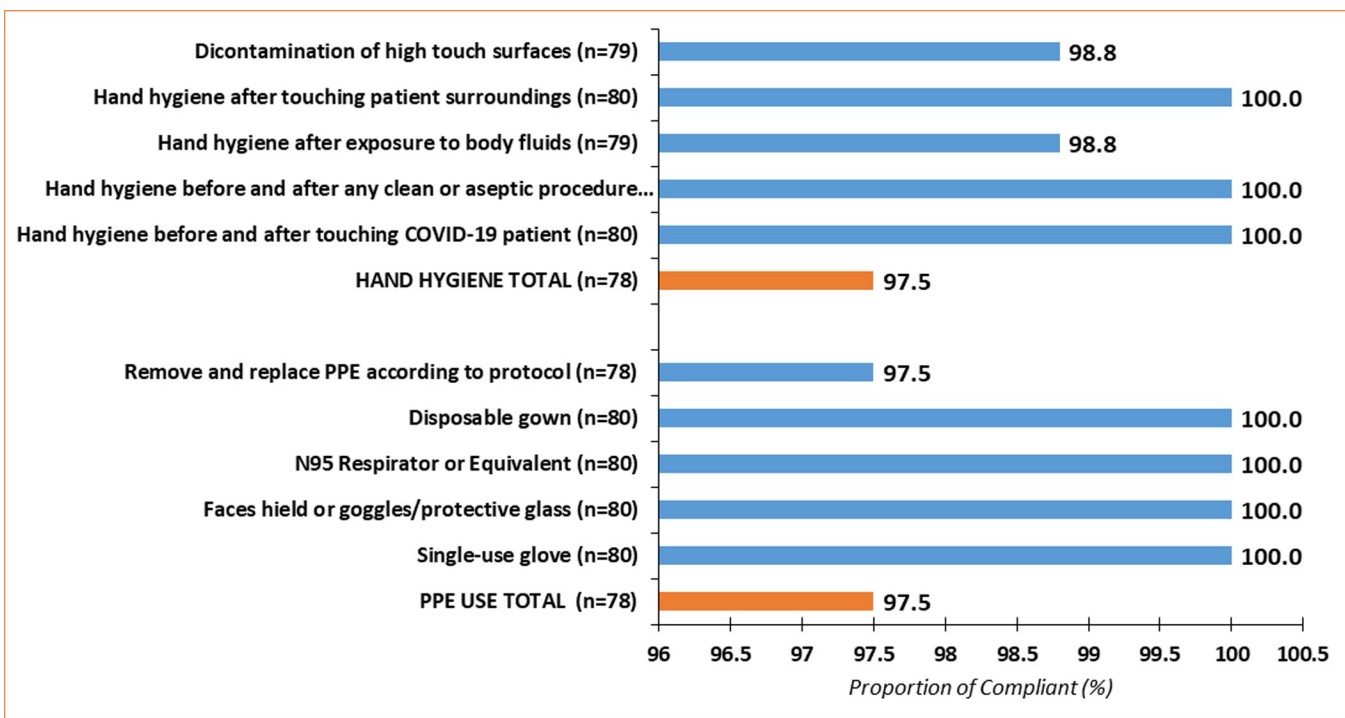

**Fig 2. Infection prevention and control reported compliance when performing aerosol-generating procedures on COVID-19 patients.**

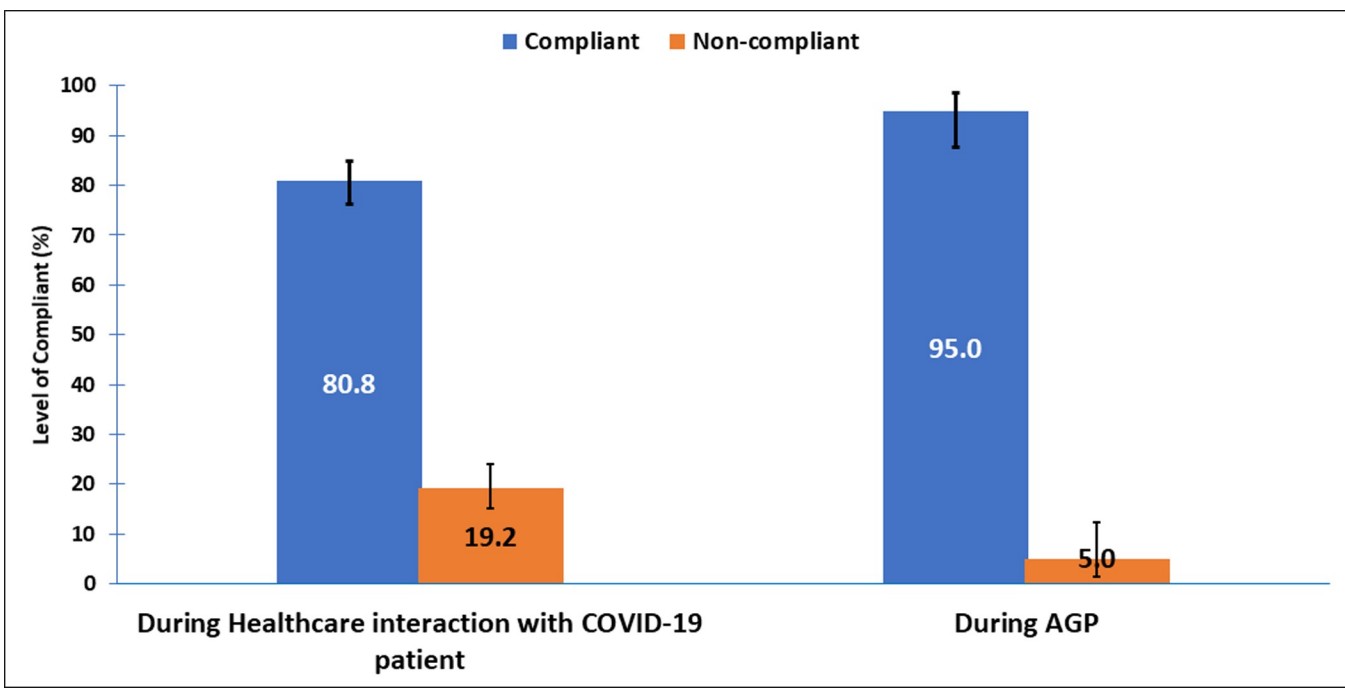

**Fig 3. Total compliance during healthcare interactions and when performing AGPs.**

pharmacists (OR 0.07; 95% CI 0.01–0.49). Insufficiency of PPEs was also associated with lower odds of compliance with PPE usage (OR: 0.33; 95% CI: 0.14–0.77).

Compliance with hand hygiene was significantly lower for healthcare workers with secondary level qualifications (OR 0.24; 95% CI 0.08–0.71) and nonclinical staff (OR 0.43; 95% CI 0.21–0.89) than healthcare workers with certificate qualifications and clinical staff. Cleaners (OR 0.27; 95% CI 0.10–0.75), midwives (OR 0.29; 95% CI 0.09–0.93), and pharmacists (OR 0.15; 95% CI 0.02–0.92) compliance with hand hygiene was significantly lower than that of registered nurses (Table 4).

## Discussion

Healthcare workers in Ghana's COVID-19 treatment centers are actively involved in managing COVID-19 cases. This put them in constant exposure to SARS-CoV-2, which can translate into COVID-19 virus infection if recommended IPC measures are not adhered to. We did a secondary analysis of healthcare workers responses on compliance with IPC measures at the COVID-19 treatment centers in Ghana. Findings from this study suggest that infection prevention and control compliance during healthcare interactions and when performing AGP was high. A high rate of IPC compliance, which is consistent with this study, has been reported in a similar study among healthcare workers [20]. However, a study among healthcare workers in Tanzanian outpatient facilities concluded that IPC compliance was inadequate [21]. The vast differences in IPC compliance may be due to the time the studies were conducted and whether compliance was observed or reported. The study in Tanzania [21] measured compliance by observation, while we measured compliance by healthcare worker self-reporting. Besides, a study in China reported improvement in IPC behaviors of healthcare workers during the COVID-19 outbreak [22]. This may be a possible explanation for high compliance with IPC protocols in this study. Infection prevention and control compliance plays a critical role

**Table 4. Association between healthcare workers' sociodemographic information, availability of IPC facilities and IPC compliance during healthcare interaction with a COVID-19 patient.**

| Healthcare worker characteristics (n = 328) | PPE Use (297/328) | | | Hand hygiene (n = 290/328) | | |
|---|---|---|---|---|---|---|
| | Compliance n/N (%) | OR [95% CI] | P-value | Compliance n/N (%) | OR [95% CI] | P-value |
| Region | | | | | | |
| Ashanti | 19/96 (20.0) | 2.30 [0.85–6.17] | 0.099 | 89/96 (92.7) | 1.96 [0.83–4.62] | 0.124 |
| Greater Accra | 57/232 (24.6) | Ref | | 201/232 (86.6) | Ref | |
| Age (In years) | | | | | | |
| < 30 | 23/124 (18.6) | 0.97 [0.44–2.13] | 0.934 | 109/124 (87.9) | 0.92 [0.46–1.85] | 0.822 |
| 30–49 | 53/204 (26.0) | Ref | | 181/204 (88.7) | Ref | |
| Gender | | | | | | |
| Female | 42/185 (22.7) | Ref | | 166/185 (89.7) | | |
| Male | 34/143 (23.8) | 0.61 [0.29–1.28] | 0.819 | 124/143 (86.7) | 0.75 [0.38–1.47] | 0.398 |
| Marital Status | | | | | | |
| Single | 33/156 (21.2) | 0.68 [0.31–1.49] | 0.379 | 136/156 (87.2) | 0.83 [0.42–1.63] | 0.583 |
| Married | 42/166 (25.3) | Ref | | 148/166 (89.2) | Ref | |
| Separated/Divorced/Widowed | 1/6 (16.7) | 0.08 [0.01–0.43] | **0.003** | 6/6 (100.0) | - | - |
| Highest Qualification | | | | | | |
| Secondary level qualification | 2/35 (5.7) | 0.24 [0.08–0.71] | **0.010** | 27/35 (77.1) | 0.24 [0.08–0.71] | **0.010** |
| Certificate | 33/107 (30.8) | Ref | | 100/107 (93.5) | Ref | |
| Diploma | 16/81 (19.8) | 0.56 [0.20–1.57] | 0.271 | 71/81 (87.7) | 0.50 [0.18–1.37] | 0.176 |
| Bachelor | 20/82 (24.4) | 1.08 [0.33–3.53] | 0.901 | 72/82 (87.8) | 0.50 [0.18–1.39] | 0.185 |
| Masters | 5/23 (21.7) | 0.74 [0.14–3.79] | 0.713 | 20/23 (87.0) | 0.57 [0.11–1.96] | 0.298 |
| Staff Category | | | | | | |
| Non-Clinical | 6/72 (8.3) | 0.16 [0.07–0.35] | **<0.001** | 58/72 (80.6) | 0.43 [0.21–0.89] | **0.021** |
| Clinical | 70/256 (27.3) | Ref | | 232/256 (90.6) | Ref | |
| Type of health professional | | | | | | |
| Assistant nurse or equivalent | 11/45 (24.4) | 0.78 [0.15–4.16] | 0.770 | 41/45 (91.1) | 0.78 [0.23–2.59] | 0.673 |
| Cleaner | 4/37 (10.8) | 0.16 [0.05–0.52] | **0.003** | 29/37 (78.4) | 0.27 [0.10–0.75] | **0.012** |
| Laboratory personnel | 6/10 (60.0) | - | - | 10/10 (100.0) | - | - |
| Medical doctor | 6/26 (23.1) | 0.91 [0.10–8.08] | 0.929 | 23/26 (88.5) | 0.58 [0.15–2.25] | 0.429 |
| Midwife | 7/24 (29.2) | 0.40 [0.07–2.18] | 0.289 | 19/24 (79.2) | 0.29 [0.09–0.93] | **0.037** |
| Registered nurse or equivalent | 38/143 (26.6) | Ref | | 133/143 (93.0) | Ref | |
| Pharmacist | 1/6 (16.7) | 0.07 [0.01–0.49] | **0.007** | 4/6 (66.7) | 0.15 [0.02–0.92] | **0.041** |
| Other staff | 3/37 (8.1) | 0.08 [0.02–0.23] | **<0.001** | 31/37 (83.8) | 0.39 [0.13–1.15] | 0.088 |
| Work Experience | | | | | | |
| < 5 | 33/167 (19.8) | Ref | | 145/167 (86.8) | Ref | |
| 5–10 | 25/104 (24.0) | 0.93 [0.40–2.15] | 0.861 | 92/104 (88.5) | 1.16 [0.55–2.46] | 0.693 |
| 11+ | 18/57 (31.6) | 0.84 [0.31–2.28] | 0.730 | 53/57 (93.0) | 2.01 [0.66–6.12] | 0.218 |
| **Availability of Infection prevention and control facilities** | | | | | | |
| Experienced interruption in water supply | | | | | | |
| No | 66/286 (23.1) | Ref | | 256/286 (89.5) | Ref | |
| Yes | 10/42 (23.8) | 2.26 [0.52–9.83] | 0.278 | 34/42 (81.0) | 0.50 [0.21–1.17] | 0.111 |
| PPE available in sufficient quantity in the health care facility | | | | | | |
| No | 13/44 (29.6) | 0.33 [0.14–0.77] | **0.010** | 37/44 (84.1) | 0.65 [0.27–1.58] | 0.339 |
| Yes | 63/284 (22.2) | Ref | | 253/284 (89.1) | Ref | |
| Received training on IPC | | | | | | |
| No | 1/7 (14.3) | 0.25 [0.05–1.34] | 0.105 | 6/7 (85.7) | 0.78 [0.09–6.67] | 0.822 |
| Yes | 75/321 (23.4) | Ref | | 284/321 (88.5) | Ref | |

in reducing healthcare workers' exposure to the COVID-19 virus. In contrast, non-compliance with IPC measures is an important factor for COVID-19 infection among healthcare workers [23]. WHO in their interim guidance on IPC recommends strict adherence to IPC protocols in managing COVID-19 patients [14].

Evidence from this study also suggests a high rate of compliance with hand hygiene protocols during healthcare interactions with COVID-19 patients. There are also reports of high hand hygiene compliance among intensive care unit healthcare workers in India [24]. However, an observational study in Turkey revealed low compliance with hand hygiene during healthcare interactions with patients [25]. Improving sustained hand hygiene compliance in healthcare settings will require continuous training of healthcare workers on IPC [24]. Additionally, the vast majority of the healthcare workers admitted that they have received training on IPC measures. This might be the reason for the high compliance reported in this study. We also found high compliance with PPE usage during healthcare interactions with COVID-19 patients by healthcare workers. This varies from a previous study conducted in Tanzania [21]. A network of factors comes into play in facilitating healthcare workers' compliance with IPC protocols. Clear IPC guidelines, effective communication, support from managers, training, access and trust in PPEs are critical in promoting healthcare compliance with IPC protocols [26]. Consistent with studies elsewhere [21, 27], there was nearly universal compliance with medical mask use during healthcare interaction with COVID-19 patients. Appropriately adhering to PPEs use is effective in reducing the risk of infection among healthcare workers [8]. Indeed, personal protective equipment use is efficacious in preventing nosocomial transmission of SARS-CoV-2 [28].

Compliance with hand hygiene and PPE usage was significantly lower among nonclinical staff than among clinical staff. This is in line with a previous study, where they found low compliance with PPEs usage among ancillary staff [29]. The risk of COVID-19 infection is not limited to only frontline healthcare workers, but other nonclinical staff, such as cleaners, drivers and security officers, also face a substantial risk of being infected with SARS-CoV-2 [30]. Perhaps, our findings may be an indication of over prioritization of IPC logistics and training of the clinical staff to the neglect of the nonclinical staff. Infection prevention and control efforts to combat the spread of COVID-19 in hospitals should include ancillary staff [30]. This is crucial in achieving zero healthcare-associated transmission of COVID-19 in healthcare settings [29].

We found the majority of the healthcare workers indicating the availability of sufficient quantities of PPEs in the treatment centers. This is in line with a previous study in Ethiopia, where most healthcare workers indicate the adequacy of infection prevention supplies in a referral hospital [31]. However, during the COVID-19 outbreak, frequent stock out and inadequate supply of PPEs have been a major challenge for healthcare workers [32] and health systems performance worldwide [33]. Ensuring the continuous availability of PPEs for healthcare workers managing COVID-19 patients is essential for maintaining healthcare workers infection rates below 10% and mortality below 1% [34]. Additionally, wide-scale procurement and distribution of PPEs for low-and-middle-income countries is cost-effective and yields a large downstream return on investment [34]. In this study, insufficiency of PPEs was associated with lower odds of compliance with PPEs usage. The role of infection prevention and control facilities in facilitating adherence with IPC measures have been reported in previous studies [31, 33, 35].

We also found lower odds of compliance with PPEs use among pharmacists compared to registered nurses. In a previous study, nurses and midwives had better compliance with glove use than other medical staffs [21]. The possible explanation for lower compliance among pharmacists may be that they do not constantly interact with patients. This could affect their compliance with PPEs usage in the treatment centers.

There was overwhelmingly high compliance with IPC protocols during AGPs by healthcare workers. Aerosol-generating procedures are high-risk procedures that are associated with an increased risk of SARS-CoV-2 transmission to healthcare workers [36]. This might be the reason for high IPC adherence among healthcare workers who performed AGPs in this study. Some AGPs generate aerosols that can facilitate the transmission of SARS-CoV-2 to healthcare workers managing COVID-19 patients. The WHO recommends the use of special respirators, gloves, aprons and eye protection during AGPs [19].

In this study, compliance with IPC protocols was self-reported by the healthcare workers, which could lead to recall bias. We could not establish how adherence to IPC protocols translates to zero COVID-19 infection in the treatment centers.

## Conclusions

Healthcare workers' compliance with IPC protocols in the treatment centers was high. The study showed wide gaps in IPC compliance across different health professional groups with nonclinical staff, cleaners, pharmacists, those with secondary level qualification and healthcare workers who report of insufficient PPEs at risk of non-compliance with IPC protocols. Ensuring an adequate supply of IPC logistics coupled with behavior change interventions and paying special attention to nonclinical staff is critical for minimizing the risk of COVID-19 transmission in the treatment centers.

## Supporting information

**S1 Table. Adherence to infection prevention and control procedures during health care interactions.**
(DOCX)

**S2 Table. Adherence to infection prevention and control measures when performing aerosol-generating procedures.**
(DOCX)

**S1 File. English questionnaire.** Questionnaire used for data collection.
(DOCX)

## Acknowledgments

We extend our gratitude to the healthcare workers and managers of the COVID-19 treatment centers.

## Author Contributions

**Conceptualization:** Mary Eyram Ashinyo, Vida Duti, Samuel Kaba Akoriyea, Patrick Kuma-Aboagye.

**Data curation:** Kingsley Ebenezer Amegah, Anthony Ashinyo, Angela Ama Ackon.

**Formal analysis:** Mary Eyram Ashinyo, Stephen Dajaan Dubik, Kingsley Ebenezer Amegah, Brian Adu Asare.

**Funding acquisition:** Mary Eyram Ashinyo.

**Investigation:** Stephen Dajaan Dubik, Vida Duti, Kingsley Ebenezer Amegah, Samuel Kaba Akoriyea.

**Methodology:** Mary Eyram Ashinyo, Stephen Dajaan Dubik, Patrick Kuma-Aboagye.

**Project administration:** Mary Eyram Ashinyo, Stephen Dajaan Dubik.

**Supervision:** Mary Eyram Ashinyo, Stephen Dajaan Dubik, Kingsley Ebenezer Amegah, Brian Adu Asare, Samuel Kaba Akoriyea.

**Visualization:** Stephen Dajaan Dubik, Kingsley Ebenezer Amegah, Angela Ama Ackon.

**Writing – original draft:** Mary Eyram Ashinyo, Stephen Dajaan Dubik, Anthony Ashinyo, Brian Adu Asare, Angela Ama Ackon.

**Writing – review & editing:** Mary Eyram Ashinyo, Stephen Dajaan Dubik, Vida Duti, Kingsley Ebenezer Amegah, Anthony Ashinyo, Brian Adu Asare, Angela Ama Ackon, Samuel Kaba Akoriyea, Patrick Kuma-Aboagye.

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
