## [Decision Letter · Decision Letter 0]

1 Feb 2021

PONE-D-20-29930

Infection Prevention and Control Compliance among Healthcare Workers in COVID-19 Treatment Centers in Ghana: A Descriptive Cross-sectional Study

PLOS ONE

Dear Dr. Ashinyo,

Thank you for submitting your manuscript to PLOS ONE. After careful consideration, we feel that it has merit but does not fully meet PLOS ONE’s publication criteria as it currently stands. Therefore, we invite you to submit a revised version of the manuscript that addresses the points raised during the review process.

We look forward to receiving your revised manuscript.

Kind regards,

Wen-Jun Tu

Academic Editor

PLOS ONE

Journal Requirements:

2. Thank you for stating in the text of your manuscript "The study was part of a large study titled “Exposure risk Assessment: A survey among frontline healthcare workers in designated COVID-19 treatment centers”, which was approved by the Ghana Health Service Ethics Review Committee. Written informed consent was obtained from all study participants.". Please also add this information to your ethics statement in the online submission form.

4. We noted in your submission details that a portion of your manuscript may have been presented or published elsewhere. ("Yes, It does not constitute a dual publication because the pending manuscripts exposure level of healthcare workers to the COVID-19 virus while the current manuscripts went ahead to measured  IPC compliance among healthcare workers who were exposed to a COVID-19 patients.) Please clarify whether this conference proceeding or publication was peer-reviewed and formally published. If this work was previously peer-reviewed and published, in the cover letter please provide the reason that this work does not constitute dual publication and should be included in the current manuscript.

5. Please ensure that you refer to Figure 3 in your text as, if accepted, production will need this reference to link the reader to the figure.

6. We note you have included tables to which you do not refer in the text of your manuscript. Please ensure that you refer to Tables 3 and 4 in your text; if accepted, production will need this reference to link the reader to the Table.

Reviewers' comments:

Reviewer's Responses to Questions

**Comments to the Author**

1. Is the manuscript technically sound, and do the data support the conclusions?

Reviewer #1: Yes

2. Has the statistical analysis been performed appropriately and rigorously? 

Reviewer #1: Yes

3. Have the authors made all data underlying the findings in their manuscript fully available?

Reviewer #1: Yes

4. Is the manuscript presented in an intelligible fashion and written in standard English?

Reviewer #1: Yes

5. Review Comments to the Author

Reviewer #1: The manuscript contains interesting and useful data about Covid-19 infection prevention and control compliance among healthcare workers in Ghana.

Minor Revision is required:

- Lines 96-98: you wrote "The sample size was calculated using the Cochran formula at a 95% confidence interval, an assumed exposure level of 50%, and a 5% margin of error."

Please quote the source, and include it in the References.

If the Cochran formula is not too long,include it as well.

- Line 117: "occasionally"

You forgot to close the quotation marks.

- Lines 306-307: "Maliszewska M, Mattoo A, Van Der Mensbrugghe D. The potential impact of COVID-19 on GDP and trade: A preliminary assessment"

Please include the year of the publication

Thank you.

6. PLOS authors have the option to publish the peer review history of their article (what does this mean?). If published, this will include your full peer review and any attached files.

Reviewer #1: **Yes: **Giovanni Vinti

---

## [Author Response · Author response to Decision Letter 0]

10 Feb 2021

To PLOS ONE Academic Editor (Wen-Jun Tu)

From the corresponding author (Dr. Mary Eyram Ashinyo)

First of all, We would like to express my sincere gratitude to you for devoting part of your schedules for our manuscript and for the valuable comments. We also express our profound gratitude to the reviewers for devoting their golden time to review this manuscript.

Below are our point by point responses to the comments raised during the review process.

Response 1: we accept this comment and we have made changes to reflect PLOS ONE style requirements

2. Thank you for stating in the text of your manuscript "The study was part of a large study titled “Exposure risk Assessment: A survey among frontline healthcare workers in designated COVID-19 treatment centers”, which was approved by the Ghana Health Service Ethics Review Committee. Written informed consent was obtained from all study participants.". Please also add this information to your ethics statement in the online submission form.

Response 2: We accept this comment and we have incorporated in the online submission form

Response 3: We accept this comment and we have incorporated the survey questionnaire as additional file “S1 File”

4. We noted in your submission details that a portion of your manuscript may have been presented or published elsewhere. ("Yes, It does not constitute a dual publication because the pending manuscripts exposure level of healthcare workers to the COVID-19 virus while the current manuscripts went ahead to measured IPC compliance among healthcare workers who were exposed to a COVID-19 patients.) Please clarify whether this conference proceeding or publication was peer-reviewed and formally published. If this work was previously peer-reviewed and published, in the cover letter please provide the reason that this work does not constitute dual publication and should be included in the current manuscript.

Response 4: We have provided the reasons for which this manuscript should not be considered as a dual publication in the cover letter. We have also included the related manuscript in this current manuscript. 

5. Please ensure that you refer to Figure 3 in your text as, if accepted, production will need this reference to link the reader to the figure.

Response 5: We accept this comment and figure 3 have been cited appropriately in the text

6. We note you have included tables to which you do not refer in the text of your manuscript. Please ensure that you refer to Tables 3 and 4 in your text; if accepted, production will need this reference to link the reader to the Table.

Response 6: We accept this comment and Table 3 and 4 have been cited accordingly in the text. 

Response 7: We accept this comment. Captions for supporting files have been updated and cited accordingly

 Lines 96-98: you wrote "The sample size was calculated using the Cochran formula at a 95% confidence interval, an assumed exposure level of 50%, and a 5% margin of error."

Please quote the source, and include it in the References.

If the Cochran formula is not too long, include it as well.

Response 8 : We accept this comment and we have reference it appropriately. We have also included the Cochran formula.

- Line 117: "occasionally"

You forgot to close the quotation marks.

Response 9: We accept this comment and quotation mark have been added accordingly

- Lines 306-307: "Maliszewska M, Mattoo A, Van Der Mensbrugghe D. The potential impact of COVID-19 on GDP and trade: A preliminary assessment"

Please include the year of the publication

Response 10: The year of publication have been incorporated appropriately.

---

## [Decision Letter · Decision Letter 1]

16 Feb 2021

PONE-D-20-29930R1

Infection prevention and control compliance among exposed healthcare workers in COVID-19 treatment centers in Ghana: a descriptive cross-sectional study

PLOS ONE

Dear Dr. Ashinyo,

Thank you for submitting your manuscript to PLOS ONE. After careful consideration, we feel that it has merit but does not fully meet PLOS ONE’s publication criteria as it currently stands. Therefore, we invite you to submit a revised version of the manuscript that addresses the points raised during the review process.

We look forward to receiving your revised manuscript.

Kind regards,

Wen-Jun Tu

Academic Editor

PLOS ONE

Additional Editor Comments (if provided):

1. In order to provide a more complete information to our readers on the topic, we would like to emphasize the importance to cross referencing very recent material on the same topic published in "PLoS ONE ". Therefore, it would be highly appreciated if you would check the contents published in the last two years of "PLoS ONE" (https://journals.plos.org/plosone/) and add all material relevant to your article to the reference list.

2. Add “Clinical Features and Short-term Outcomes of 102 Patients with Corona Virus Disease 2019 in Wuhan, China. Clinical Infectious Diseases, 71(15):748-755” in the revision text。

3. The grammar needs to be edited throughout.

Reviewers' comments:

Reviewer's Responses to Questions

**Comments to the Author**

1. If the authors have adequately addressed your comments raised in a previous round of review and you feel that this manuscript is now acceptable for publication, you may indicate that here to bypass the “Comments to the Author” section, enter your conflict of interest statement in the “Confidential to Editor” section, and submit your "Accept" recommendation.

Reviewer #1: (No Response)

2. Is the manuscript technically sound, and do the data support the conclusions?

Reviewer #1: Yes

3. Has the statistical analysis been performed appropriately and rigorously? 

Reviewer #1: Yes

4. Have the authors made all data underlying the findings in their manuscript fully available?

Reviewer #1: Yes

5. Is the manuscript presented in an intelligible fashion and written in standard English?

Reviewer #1: Yes

6. Review Comments to the Author

Reviewer #1: Lines 78-80:

Please, mention Covid-19 vaccines as well. For example, here you can find some clues:

- Matrajt et al., 2021. Vaccine optimization for COVID-19: Who to vaccinate first? Science Advances Vol. 7, no. 6, eabf1374. DOI: 10.1126/sciadv.abf1374

Line 93: “formula [12];” and “with the following parameters;”

You should use “:” instead of “;”.

Lines 258-169: I did not understand how you obtained 23.2% for “PPE use total”. Could you please recheck it and clarify it in the text? Because it seems very low, compared to the other individual values you showed, and even checking Supporting information – S1 file and S1 table, it is not clear to me how you got that value. Therefore, a better explanation would be needed.

7. PLOS authors have the option to publish the peer review history of their article (what does this mean?). If published, this will include your full peer review and any attached files.

Reviewer #1: **Yes: **Giovanni Vinti

---

## [Author Response · Author response to Decision Letter 1]

19 Feb 2021

1. In order to provide a more complete information to our readers on the topic, we would like to emphasize the importance to cross referencing very recent material on the same topic published in "PLoS ONE ". Therefore, it would be highly appreciated if you would check the contents published in the last two years of "PLoS ONE" (https://journals.plos.org/plosone/) and add all material relevant to your article to the reference list.

We accept this recommendation, but we are unable to add references that cannot be cited in the text. We have however, added articles published in PLOS ONE which are citable in our work. References 9, 11, 13, 32, 34 and 36

2. Add “Clinical Features and Short-term Outcomes of 102 Patients with Corona Virus Disease 2019 in Wuhan, China. Clinical Infectious Diseases, 71(15):748-755” in the revision text。

We accept this recommendation and we have cited it in the manuscript, reference 6

3. The grammar needs to be edited throughout.

The manuscript have undergone grammar editing

Reviewer #1: Lines 78-80:

Please, mention Covid-19 vaccines as well. For example, here you can find some clues:

- Matrajt et al., 2021. Vaccine optimization for COVID-19: Who to vaccinate first? Science Advances Vol. 7, no. 6, eabf1374. DOI: 10.1126/sciadv.abf1374

We accept this comment and we have incorporated it accordingly. Reference 16; lines 83-84

Line 93: “formula [12];” and “with the following parameters;”

You should use “:” instead of “;”.

we have modified it to suit this comment, but we rather used “at” 95% confidence interval (1.96)

Lines 258-169: I did not understand how you obtained 23.2% for “PPE use total”. Could you please recheck it and clarify it in the text? Because it seems very low, compared to the other individual values you showed, and even checking Supporting information – S1 file and S1 table, it is not clear to me how you got that value. Therefore, a better explanation would be needed.

Thank you very much for this important comment. We realized that there was a mistake in recoding some of the variables into compliant and non-compliant. This has been rectify accordingly. How compliant and non-compliant was measure have been explained in Table 1 and 2.

---

## [Editor Report · Decision Letter 2]

24 Feb 2021

Infection prevention and control compliance among exposed healthcare workers in COVID-19 treatment centers in Ghana: a descriptive cross-sectional study

PONE-D-20-29930R2

Dear Dr. Ashinyo,

We’re pleased to inform you that your manuscript has been judged scientifically suitable for publication and will be formally accepted for publication once it meets all outstanding technical requirements.

Kind regards,

Wen-Jun Tu

Academic Editor

PLOS ONE
---

## [Editor Report · Acceptance letter]

26 Feb 2021

PONE-D-20-29930R2 

Infection prevention and control compliance among exposed healthcare workers in COVID-19 treatment centers in Ghana: a descriptive cross-sectional study 

Dear Dr. Ashinyo:

I'm pleased to inform you that your manuscript has been deemed suitable for publication in PLOS ONE. Congratulations! Your manuscript is now with our production department. 

Kind regards, 

on behalf of

Dr. Wen-Jun Tu 

Academic Editor

PLOS ONE